# Efficacy of propylthiouracil in the treatment of pregnancy with hyperthyroidism and its effect on pregnancy outcomes: A meta-analysis

Yiqun Miao[1], Yang Xu[1], Ping Teng[2], Aihua Wang[1]*, Yuanyuan Zhang[1], Yun Zhou[1], Wenwen Liu[1]

1 School of Nursing, Weifang Medical University, Weifang, Shandong, China, 2 Delivery Room, Weifang Maternal and Child Health Care Hospital, Weifang, Shandong, China

* wangaihua64@163.com

**Data Availability Statement:** All relevant data are within the manuscript and its Supporting Information files.

## Abstract

### Background

Hyperthyroidism affects about 0.2%-2.7% of all pregnancies, and is generally treated with propylthiouracil (PTU). However, previous studies about the effects of propylthiouracil on maternal or foetal are contentious.

### Objective

This meta-analysis was carried out to investigate the safety and efficacy of propylthiouracil during pregnancy.

### Materials and methods

PubMed, EBSCO, Embase, Scopus, Web of Science, Cochrane, CNKI, Wanfang and VIP database were searched from inception until August 31, 2021 for all available randomized controlled trials (RCTs) or cohort studies that evaluated the efficacy of propylthiouracil and its effects on pregnancy outcomes. Odds ratio (OR) and 95% confidence interval (CI) were used for binary variables, weighted mean difference (WMD) and 95% confidence interval (CI) were used for continuous variables. RevMan5.4 and Stata 16.0 were used for performing the meta-analysis.

### Results

The researchers examined data from 13 randomized controlled trials and cohort studies involving 18948 infants. Congenital anomalies were not significantly associated with PTU in the pooled results (OR = 1.03, 95%CI: 0.84–1.25, $P$ = 0.80, $I^2$ = 40.3%). There were no statistically significant differences in neonatal hypothyroidism (OR = 0.55, 95%CI: 0.06–4.92, $P$ = 0.593, $I^2$ = 57.0%) or hepatotoxicity (OR = 0.34, 95%CI: 0.08–1.48, $P$ = 0.151, $I^2$ = 0.0%) exposed to PTU compared to the control group. The serum levels of FT3, FT4, TT3, and TT4 were significantly lower in the propylthiouracil group compared to the control group.

**Funding:** The author(s) received no specific funding for this work.

**Competing interests:** The authors have declared that no competing interests exist.

**Abbreviations:** PTU, propylthiouracil.

## Conclusion

This meta-analysis confirmed the beneficial effects of propylthiouracil treatment, namely the risks of adverse pregnancy outcomes were not increased, and it also proved PTU's efficacy in the treatment of pregnant women with hyperthyroidism. The findings supported the use of propylthiouracil during pregnancy with hyperthyroidism in order to improve clinical pregnancy outcomes in patients with thyroid dysfunction.

## Introduction

Pregnancy hyperthyroidism is one of the most common pregnancy complications, affecting about 0.2–2.7 percent of pregnancies and having an incidence rate of about 65 per 100,000 [1–3]. Graves' disease and chorionic gonadotrophin (hCG)-mediated hyperthyroidism are the two most common causes of hyperthyroidism during pregnancy [4]. Untreated disease increases the risk of miscarriage, premature delivery, placental abruption, stillbirth, and other complications [5]. Therefore, appropriate management of hyperthyroidism during pregnancy is critical. Clinical guidelines recommending treatment parallels that propylthiouracil should be used as the first line treatment of hyperthyroidism in pregnancy [6, 7]. However, the drug is regarded to pass through the human placenta, it may have an effect on the foetus [8].

Several observational studies have been conducted to investigate the effects of propylthiouracil on the occurrence of adverse pregnancy outcomes, but the findings have been inconclusive and controversial. There is no comprehensive data on the adverse pregnancy outcomes associated with PTU. Several studies have shown that propylthiouracil is hepatotoxic [9, 10], plus research has found that the drug may have teratogenic effects as well [11, 12]. According to a recent cohort study, PTU could lead to neonatal hypothyroidism with or without goitre [13]. In contrast, some studies found no adverse effects of PTU on pregnancy outcomes when conducted retrospectively [14, 15].

As a result, no final conclusion on the use of PTU has been reached, and more research is needed, particularly to investigate the underlying role of propylthiouracil in maternal or foetal health. Therefore, it is necessary to reassess previously published research evidence on the role of PTU treatment in the risk of pregnancy outcomes and its efficacy. Combining and analyzing data from this contentious issue may provide clinical physicians with medication guidelines for pregnant women with hyperthyroidism.

## Materials and methods

### Literature search

The meta-analysis followed the statement of the Preferred Reporting Items for Systematic Reviews and Meta-analyses (PRISMA) [16]. And the following databases were searched: CNKI, Pubmed, Cochrance, Embase, Scopus, Web of Science, Wanfang and VIP database. These articles were published after inception and before August 2021, which were related to the effectiveness and safety of propylthiouracil treatment on pregnant women. Comparisons with untreated control groups were conducted.

The authors searched independently before the study started for the determination of inclusion criteria. On the basis of MESH, Pubmed was searched first with below keywords: ("pregnancy" OR "Pregnancies" OR "Gestation") AND ("hyperthyroidism" OR "Hyperthyroid" OR "Hyperthyroids" OR "Primary Hyperthyroidism" OR "Hyperthyroidism, Primary"") AND

("Propylthiouracil" OR "6-Propyl-2-Thiouracil" OR "6 Propyl 2 Thiouracil"). Only studies involving humans were included for selection. Moreover, language limitations are English or Chinese. Similar searching strategies were applied in all databases. Searches were performed based on "all fields" in the PubMed and on "titles, abstracts and keywords" in other databases.

## Eligibility criteria

**Types of participants.**    In this study, the selected participants needed to satisfy the following criteria: a study population including pregnant women who are clinically and laboratory diagnosed with hyperthyroidism. Hyperthyroidism refers to high levels of serum thyroxine and triiodothyronine, and low levels of thyroid-stimulating hormone [17].

**Intervention measures.**    The control group received no treatment.

The experimental group received propylthiouracil treatment.

**Outcomes.**    The primary outcomes:

Congenital anomalies refer to structural or functional defects occurring in prenatal development [18], classified per International Classification of Diseases, 10th Revision [ICD-10], codes Q00 to Q99.

Neonatal thyroid hypothyroidism is defined as abnormal indicators of thyroid function such as thyroid volume (TV), thyroid-stimulating hormone (TSH), T3 and T4.

Hepatotoxicity: Abnormal Aspartate Aminotransferase (AST) and Alanine Aminotransferase (ALT) were used as indicators of hepatotoxicity [19].

The secondary outcomes:

Thyroid function in pregnant women: free triiodothyronine (FT3), total triiodothyronine (TT3), free thyroxine (FT4), and total thyroxine (TT4) concentrations.

Free triiodothyronine (FT3), free thyroxine (FT4) were measured by an electrochemiluminescence immunoassay on a Cobas Elecsys 601 unit (Roche Diagnostics, Basel, Switzerland). Total thyroxine(TT4) and total triiodothyronine (TT3) were measured using radioimmunoassay. The FT4 and FT3 reference intervals were 9.2–21.0 pmol/L and 3.52–5.61pmol/L; TT4 was 86.9–213.2nmol/L and TT3 was 1.32–3.72 nmol/L in pregnancy [20].

## Exclusion criteria

For the meta-analysis, any study that meets a certain exclusion criterion must be excluded in this paper: (1) Literature with incomplete reporting information for data. (2) Literature without adequate data for result interpretation. (3) Literature that including other anti-thyroid therapies instead of propylthiouracil. (4) Review, studies related to animal trial, in vitro study, case report, meeting abstract.

## Data extraction and quality assessment

Two reviewers independently performed the research screening. After the databases search were finished, all the filtered titles and abstracts of literature were imported into the document management software for duplicate studies screening. Those failing to satisfy the inclusion criteria were screened, and eventually the studies that meet the inclusion criteria were sorted out through full text reading. In literature screening, if disagreement arose, reviewers should discuss with a third researcher or consult experts relevant to the research field. In case of various papers having been published in the same research, only the research that has the most complete data and the most coincident with the inclusion criteria should be included. All the selected studies satisfying the inclusion criteria were selected and analyzed using computer: basic research information (author, title, study type, year of publication), experimental design (number of cases, dosage, interventions,

duration of exposure) and outcome indicators. If there were inconsistent outcome indicator units in the study, the units should be converted in uniform before carrying out subsequent data processing. If some information was missing in the research, the original author should be contacted by telephone or e-mail for relevant data.

Cochrane Handbook recommended a method for System Reviews of Interventions 5.1.0, the evaluation items of which include random grouping methods, use of blind methods, allocation of hidden scheme design, reporting of results data, whether there are other sources of bias, whether there is any selective reporting of research results, etc. The selected studies were evaluated for bias risk using the method and its results were reported as follows: "low risk" represents correct method or the complete data, illustrating small bias risk; "Unclear" means unclear method, illustrating moderate bias risk; "high risk" indicates that incorrect method or incomplete data, illustrating high bias risk. Finally, the bias risk assessment chart could be obtained after the evaluation results were input into RevMan 5.4 software.

### Statistical analysis

To analyze statistics, the Software package STATA (version 16; STATA Inc., College Station, TX, USA) was utilized. The weighted mean differences (WMD) were used to compare continuous variables, while categorical data utilized by pooled odds ratios (ORs) were obtained to determine whether there were significant differences in the compared data. Confidence interval (CI) of 95% was used to express the effect of the quantity of counting data and the measurement data.

In the heterogeneity test, in case of the statistics $P > 0.1$, $I^2 < 50\%$ was generally regarded as an indicator of a higher homogeneity among study results, revealing no significant statistical difference in the included data. Under the circumstances, the fixed effect model should be employed. In case of statistic $P \leq 0.1$, $I^2 \geq 50\%$ was generally regarded as an indicator of heterogeneity among study results, indicating significant statistical differences in the inclusion data, and the random effect models were used. And given the factors that might result in heterogeneity, heterogeneity probably exists. If a study varied significantly from all other included studies in methods or results, a sensitivity analysis would be conducted to exclude those studies from the meta-analysis. Funnel plots were generated and the Begg's and Egger's test were employed to assess and test publication bias, respectively. $P < 0.05$ indicated statistical significance, unless otherwise specified.

## Results

### Search results

S1 Materials shows the search strategy in detail. Fig 1 shows the PRISMA flow chart for literature selection and the PRISMA checklist is shown in the S1 Table. There were totally 1912 studies in the nine databases (PubMed: 430; EBSCO: 198; Embase: 707; Web of science: 145; Cochrane: 6; Scopus: 340; CNKI: 49; wanfang database: 24; VIP: 13). There were no additional records from other sources found. After removing duplicates, up to 1090 records remained, and 820 of them were delegated after the titles and abstracts were examined. In addition, full texts of 35 articles were read, with 22 of them being refused. Finally, 13 studies were included in the meta-analysis [14, 15, 21–31].

### Study characteristics and quality assessment

This meta-analysis included 13 articles. The experiment group included pregnant women who accepted propylthiouracil treatment for a normal range of thyroid hormone levels during

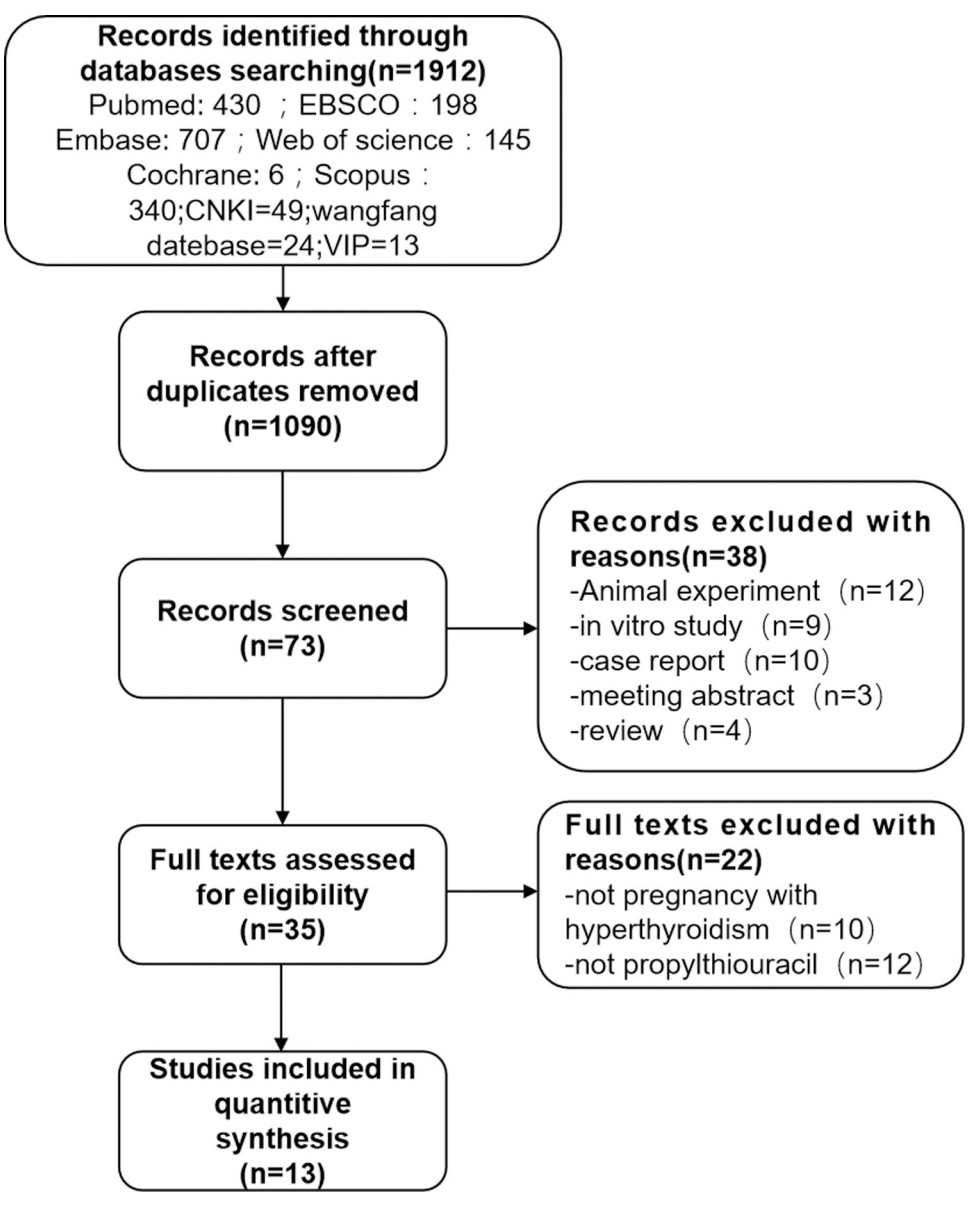

**Fig 1. Flow chart of study selection process.**

hyperthyroidism. The control group included pregnant women with hyperthyroidism who had not received antithyroid treatment prior to delivery. Table 1 shows the features of articles included in this study.

Figs 2 and 3 depict the risk-of-bias assessment in minute detail. Only two studies demonstrated a low risk of bias [21, 25], and an unclear risk of bias was found in eleven studies [14, 15, 22–24, 26–31]. Six studies' randomization was handled correctly [21, 22, 25, 28–30], and five studies dealt with allocation-sequence concealment adequately. Details about participants and personnel blinding were provided in nine studies [14, 15, 21–27], while outcome assessor blinding was reported in nine more [14, 15, 21, 24, 25, 27, 29–31]. For each article, the cause and number of withdrawals and dropouts were provided.

**Table 1. Characteristics of included studies.**

| Study | Year | Study type | Participants | Controls | Duration of exposure | Outcomes |
|---|---|---|---|---|---|---|
| Hao | 2018 | RCT | 56 women treated with PTU; The initial dose is 100mg, three times a day, and the dosage is increased or decreased according to the disease | 122 women not receiving any treatment | Before the end of 18 weeks of gestation | ① |
| Andersen | 2017 | Cohort study | 218 women treated with PTU; Dose unknown | 1551 women not receiving any treatment | During the first trimester of pregnancy | ① |
| Ji | 2017 | RCT | 20 women treated with PTU; The initial dose is 100mg, three times a day, and the dosage is increased or decreased according to the disease | 20 women not receiving any treatment | During pregnancy | ③④⑤⑥⑦ |
| Lo | 2015 | Cohort study | 433 women treated with PTU; Dose unknown | 1144 women not receiving any treatment | Before the end of 2 months of gestation | ①② |
| Gianetti | 2015 | Cohort study | 52 women treated with PTU; daily dose ranges from 50 to 200 mg | 203 women not receiving any treatment | During the first trimester of pregnancy | ①③ |
| Andersen | 2014 | Cohort study | 564 women treated with PTU; Dose unknown | 3543 women not receiving any treatment | During the first trimester of pregnancy | ① |
| Guo | 2014 | RCT | 58 women treated with PTU; The initial dose is 100mg, three times a day, and the dosage is increased or decreased according to the disease | 52 women not receiving any treatment | During pregnancy | ③④⑤⑥⑦ |
| Xu | 2013 | RCT | 65 women treated with PTU; The initial dose is 100mg, three times a day, and the dosage is increased or decreased according to the disease | 65 women not receiving any treatment | During the first trimester of pregnancy | ①④⑤⑥⑦ |
| Korelitz | 2013 | Cohort study | 915 women treated with PTU; Dose unknown | 3236 women not receiving any treatment | Before the end of 6 months of gestation | ①② |
| Yoshihara | 2012 | Cohort study | 1578 women treated with PTU; Dose unknown | 2065 women not receiving any treatment | During the first trimester of pregnancy | ① |
| Chen | 2011 | Case-control | 630 women treated with PTU; Dose unknown | 2127 women not receiving any treatment | During the first trimester of pregnancy | ① |
| Lian | 2005 | Case-control | 28 women treated with PTU; Dose unknown | 61 women not receiving any treatment | During the first trimester of pregnancy | ① |
| Wing | 1994 | Cohort study | 99 women treated with PTU; The median maximal daily medication dose of propylthiouracil was 450 mg with a range of 150 to 600mg | 43 women not receiving any treatment | Before the end of 15 weeks of gestation | ①③ |

Note: ①Congenital anomalies ②hepatotoxicity ③infants thyroid dysfunction ④FT3 ⑤FT4 ⑥TT3 ⑦TT4

## Meta analysis results

Many studies have been conducted to assess the risk of congenital anomalies, neonatal hypothyroidism, and hepatotoxicity in pregnant women exposed to propylthiouracil, but the associations are unclear. The current findings provide greater power to the results of pregnancy outcomes for PTU treatment based on 13 RCTs and cohort studies.

**Congenital anomalies.** Eleven studies with a total of 18460 infants reported findings on congenital anomalies. Because heterogeneity analysis revealed that the included studies were

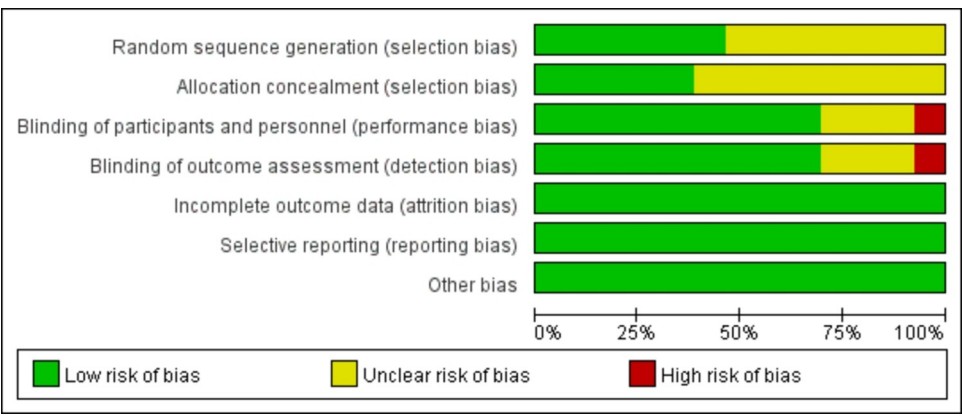

**Fig 2. Risk of bias assessment.**

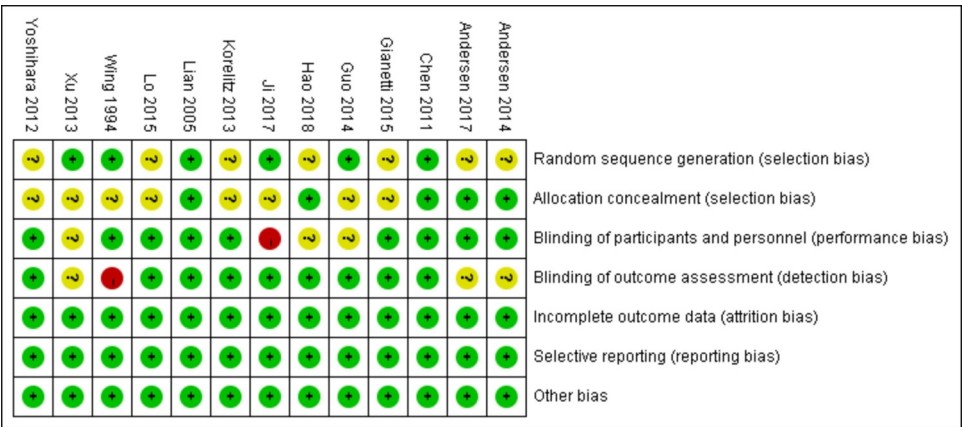

**Fig 3. Trial quality assessment.**

highly homogeneous ($I^2$ = 40.3%, $P$ = 0.08), the fixed effect model was chosen, and sensitivity analysis demonstrated that the results were reliable. The meta-analysis of these eleven studies on the association of propylthiouracil treatment with congenital malformations yielded a pooled OR of 1.03, with a 95% CI of 0.84–1.25, demonstrating no significant differences in this outcome between the treatment group and control group ($P$ = 0.80). This finding illustrated that propylthiouracil did not pose a significant teratogenic risk (Fig 4).

**Neonatal hypothyroidism.** Four studies involving 547 infants reported findings on neonatal hypothyroidism. The heterogeneity test for these outcomes was significant ($I^2$ = 57.0%, $P$ = 0.073), so the random effects model was adopted. There were no statistically significant differences in the pooled odds ratios of neonatal hypothyroidism between women treated with

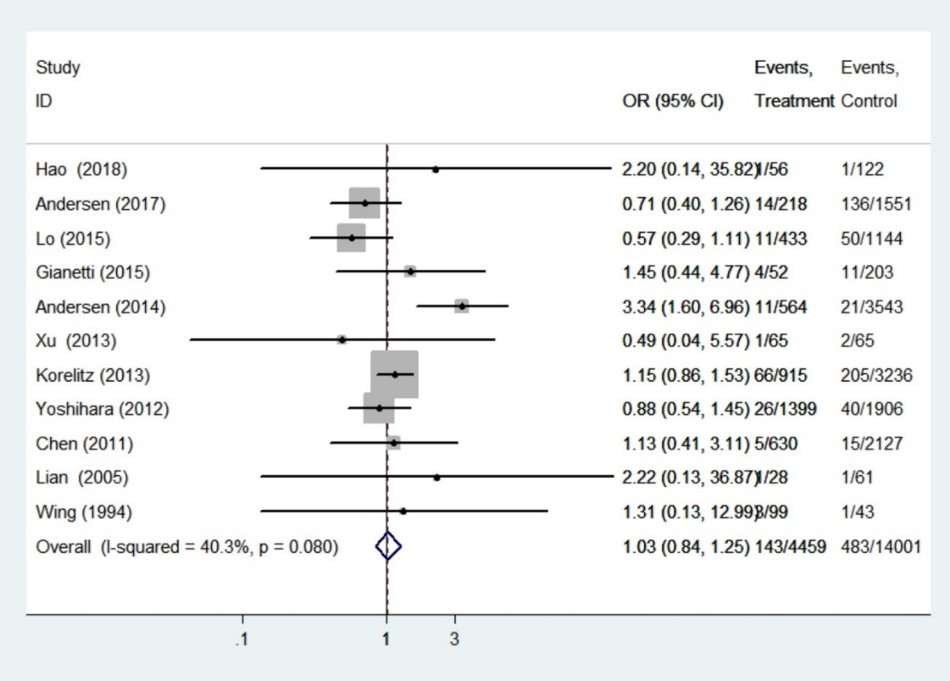

**Fig 4. Forest plots of showing the effects of PTU on congenital anomalies.**

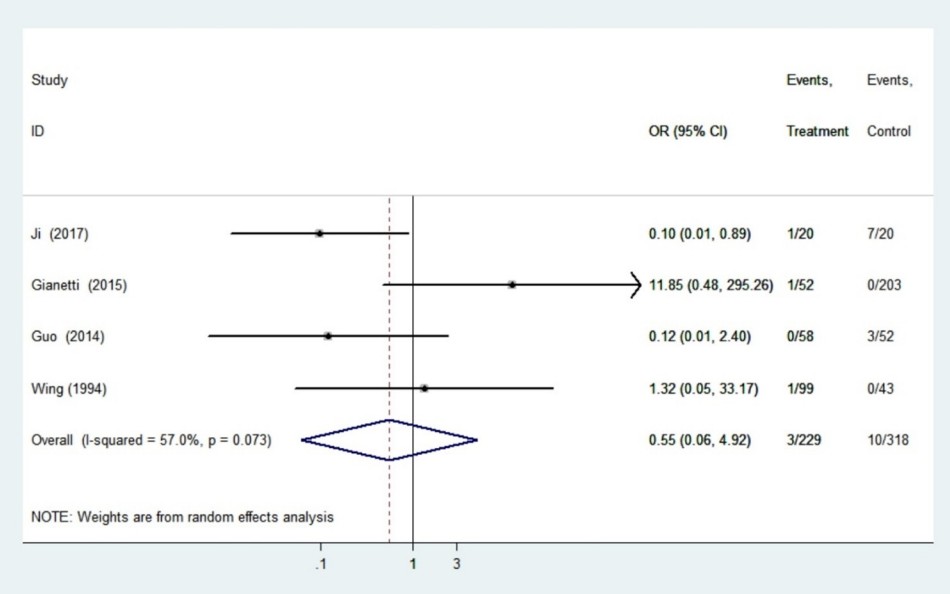

**Fig 5. Forest plots of showing the effects of PTU on neonatal hypothyroidism.**

PTU and the control group (OR 0.55, 95%CI 0.06–4.92, $P = 0.593$). Sensitivity analyses using sequential omission of individual studies had no significant effect on the overall combined OR, indicating that the combined OR was credible and valid. We found no link between the use of propylthiouracil in pregnancy and the risk of neonatal thyroid dysfunction in women with hyperthyroidism (Fig 5).

**Hepatotoxicity.**   Two studies with a total of 8012 patients reported hepatotoxicity findings. There was no statistical heterogeneity among the research ($I^2 = 0.0\%$, $P = 0.959$), so the fixed effect model was used. Meta-analysis showed that there was no significant difference in hepatotoxicity between propylthiouracil treatment group and the control group in pregnant women with hyperthyroidism (OR 0.34, 95%CI: 0.08–1.48, $P = 0.151$) (Fig 6). We found no evidence of an increased risk of liver disease in women who took propylthiouracil.

To explore the efficacy of the propylthiouracil in pregnant women, we also analyzed the values of FT3, FT4, TT3 and TT4 at the same time.

**FT3.**   Three studies with 215 patients reported results on FT3. There was no statistical heterogeneity among the studies ($I^2 = 0.0\%$, $P = 0.552$), so the fixed effect model was adopted. In women with hyperthyroidism, the value of FT3 was significantly decreased in propylthiouracil treatment than the control group(WMD:-14.09,95%CI: -14,51– -13.68, $P<0.001$) (Fig 7). Propylthiouracil achieved a stable FT3 state in patients, according to the findings.

**FT4.**   Three studies with a total of 215 patients reported FT4 results. There was a little statistical heterogeneity among the studies ($I^2 = 55.3\%$, $P = 0.107$), so random effects model was adopted. All of these studies concluded that propylthiouracil significantly reduced the value of FT4 in hyperthyroid women (WMD: -53.98, 95%CI: -56.06–-51.87, $P<0.001$). Our study revealed that propylthiouracil could improve thyroid function in pregnant women with hyperthyroidism (Fig 8).

**TT3.**   A total of two studies were included in this study, and there was low statistical heterogeneity among the studies ($I^2 = 0.0\%$, $P = 0.623$). Therefore, the fixed effect model was used

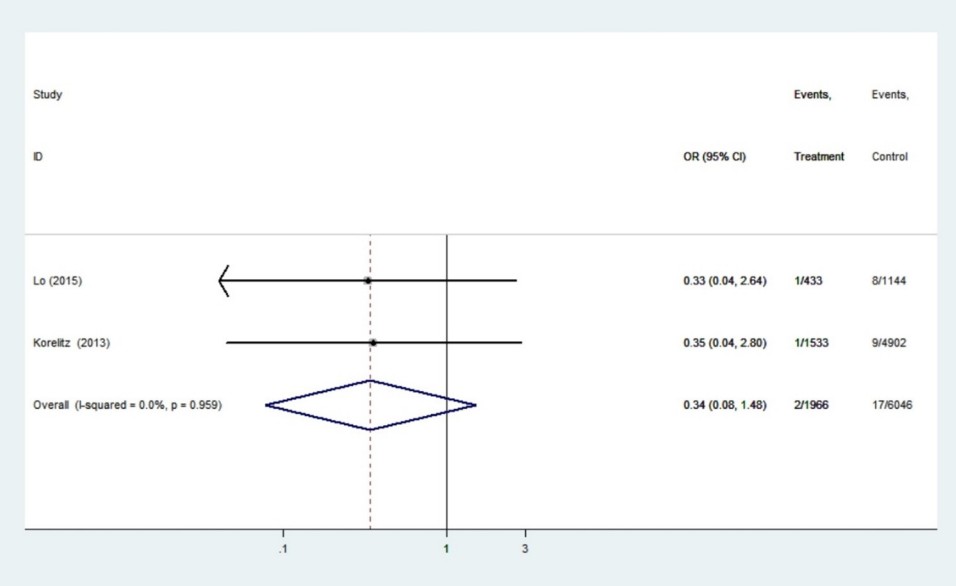

**Fig 6. Forest plots of showing the effects of PTU on hepatotoxicity.**

for meta-analysis, and it seemed that the difference in TT3 between the treatment group and the control group was statistically significant. In patients with hyperthyroidism, the value of TT3 was significantly decreased in the experimental group vs. the control group (WMD: -1.45,95%CI: -1.73– -1.17, $P<0.001$) (Fig 9). Propylthiouracil was found to be effective in the treatment of hyperthyroidism.

**TT4.** A total of 3 studies were included in this study, and the statistical heterogeneity among literature was small ($I^2$ = 38.1%, $P$ = 0.199). Therefore, the fixed effect model was used for meta-analysis. We did find a significant difference in the value of TT4 (WMD: -153.67,

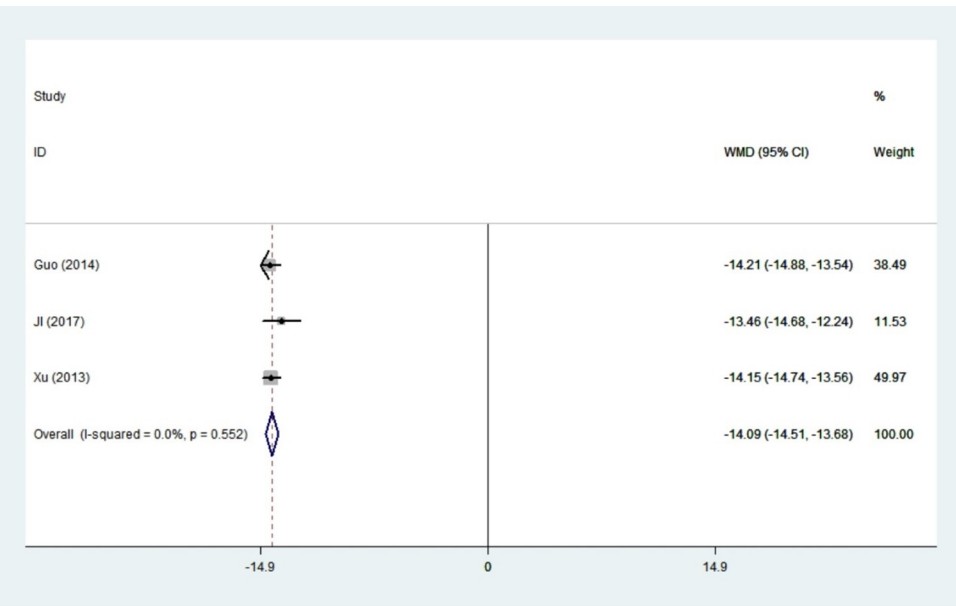

**Fig 7. Forest plots of showing the effects of PTU on FT3.**

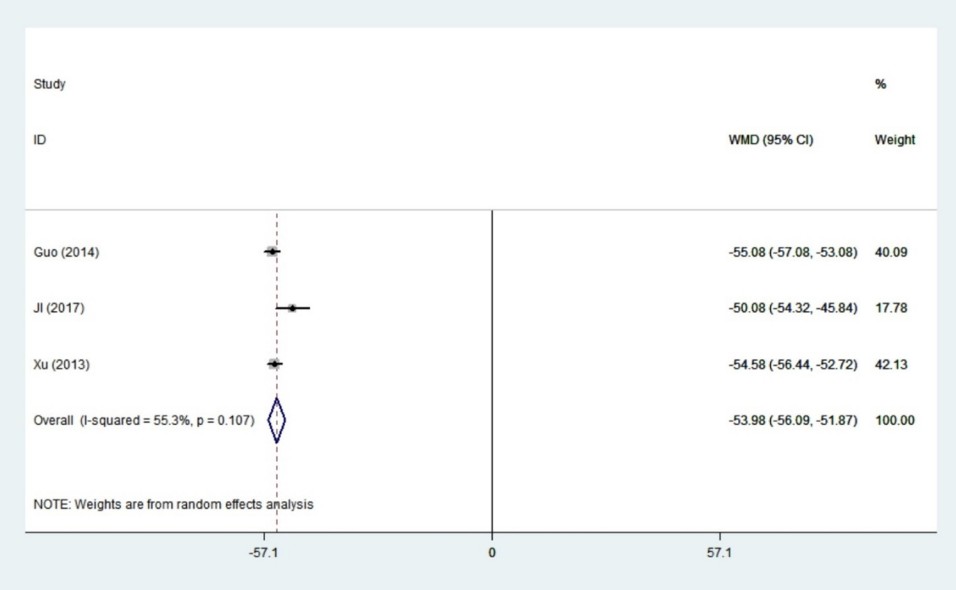

**Fig 8. Forest plots of showing the effects of PTU on FT4.**

95%CI: -160.73–-146.62, $P<0.001$) (Fig 10). The combined results of these studies revealed that propylthiouracil treatment had a remarkable effect on the TT4.

## Publication bias

To assess the study's quality and risk of bias, various complementary methods were used, including funnel plots, Begg's and Egger's tests. In studies on hyperthyroid patients, there was a clear symmetry funnel plot for congenital anomalies (Fig 11). The results of Egger's ($P = 0.768$) and Begg's ($P = 0.64$) tests also confirmed that there was no significant risk of bias in the study.

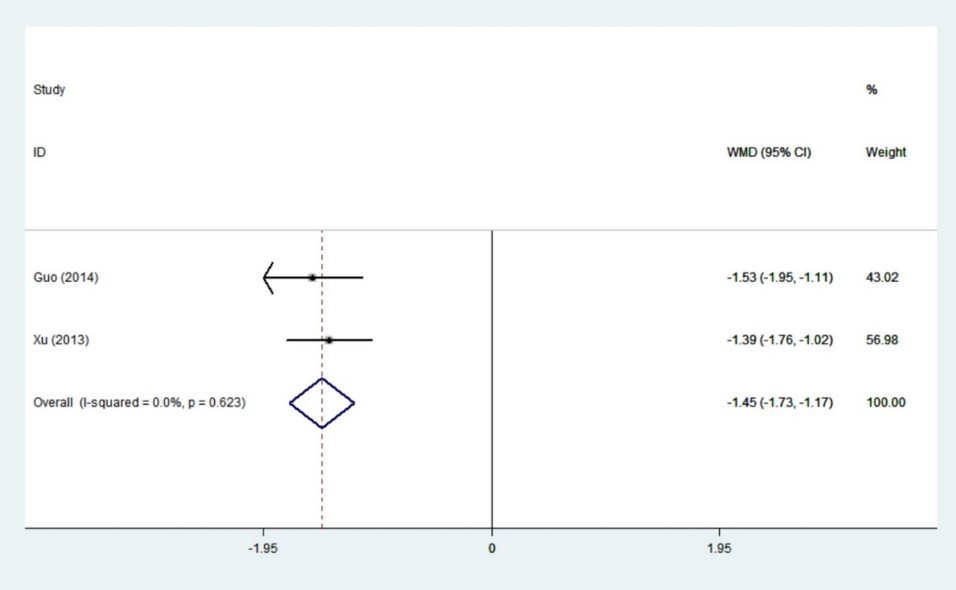

**Fig 9. Forest plots of showing the effects of PTU on TT3.**

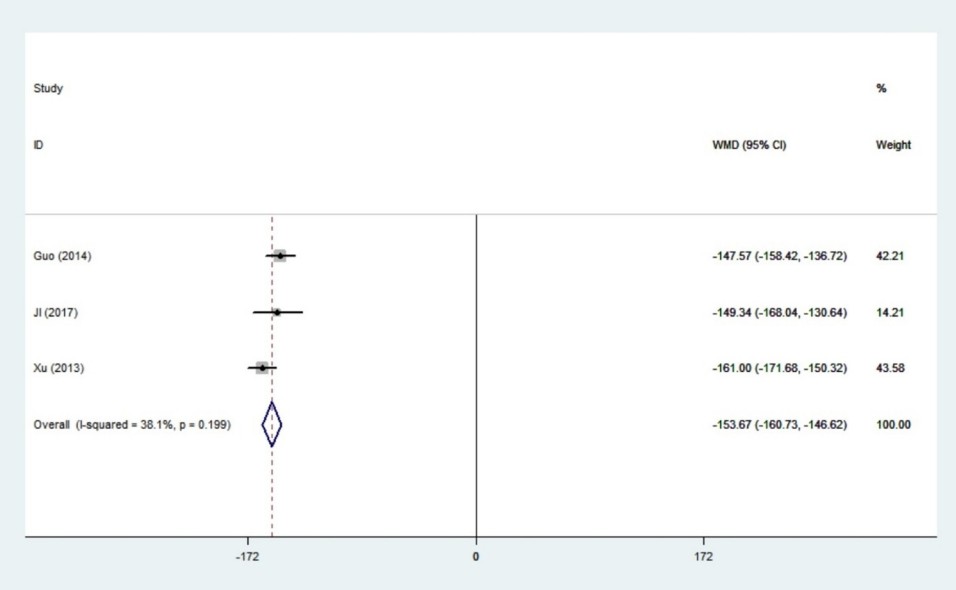

**Fig 10. Forest plots of showing the effects of PTU on TT4.**

## Discussion

Thyroid hormone is essential for normal pregnancy and intrauterine foetal development, particularly growth of the fetal brain [32, 33]. There is compelling evidence that pregnant women should be treated for hyperthyroidism [34, 35]. However, the therapy for hyperthyroidism is limited because all available treatments have reported adverse effects on pregnancy. The option

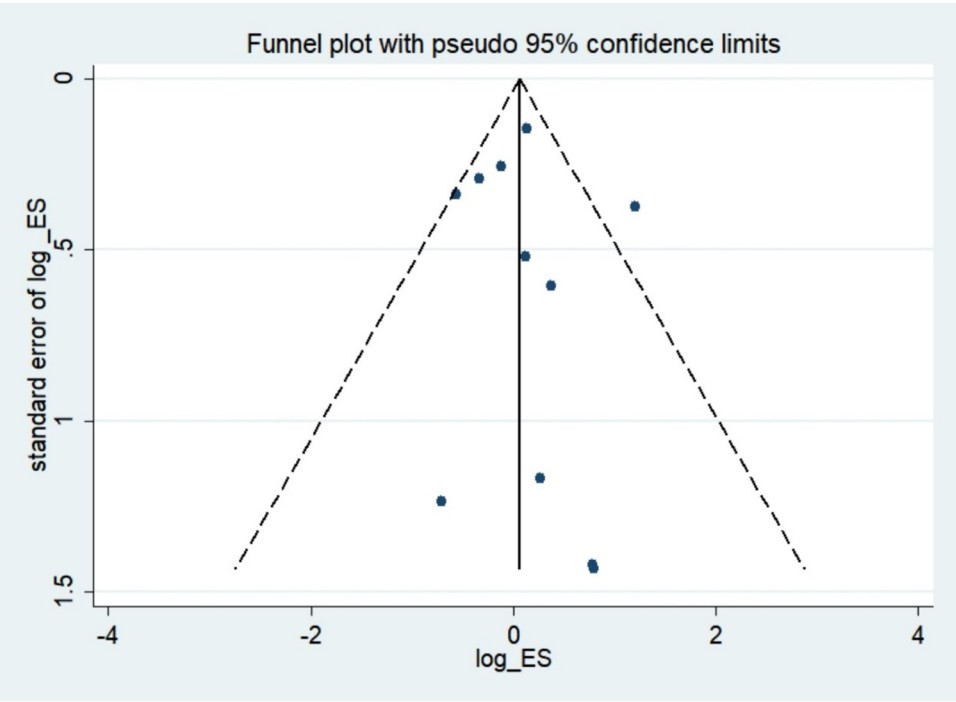

**Fig 11. Funnel plot of congenital anomalies.**

is ultimately dependent on the balance of undesirable side effect burden associated with these medications. PTU is commonly used in the first trimester of pregnancy, but the risks of propylthiouracil-related pregnancy outcomes are debatable.

In this report, we utilized meta-analysis for the first time to summarize the pooled ORs of pregnancy outcomes and efficacy after exposure to propylthiouracil in pregnant women. Concerning the safety profile, we discovered that hyperthyroidism patients treated with PTU had no significant differences in the OR of adverse pregnancy outcomes, such as congenital anomalies, hepatotoxicity, and neonatal thyroid hypothyroidism, when compared to no treatment controls, indicating that propylthiouracil therapy did not increase the risk of unwanted side effects on pregnancy outcomes. The study also discovered the effectiveness of propylthiouracil on the therapy among pregnant women with hyperthyroidism. Further research revealed that propylthiouracil may significantly lower serum levels of FT4, FT3, TT4, and TT3.

The current study team analyzed the effect sizes of all included studies and discovered that PTU had no relationship with congenital anomalies. In the present study, ten of the eleven studies that evaluated the relationship between propylthiouracil and the risk of congenital anomalies found no link. Our findings were consistent with previous systematic reviews [36–38], which found that the risk of congenital anomalies was similar in hyperthyroid women treated with PTU and those who did not accept any antithyroid drug. Previous research by Li, H et al. [39] suggested that the risk of birth defects increased with PTU exposure, which contradicted our findings. Perhaps the reason was that the subjects in their control group were healthy women. It is possible that the risk of propylthiouracil-induced anomalies is higher than in women without hyperthyroidism, but not higher than the risk in hyperthyroid patients who do not receive treatment. Furthermore, due to its high molecular weight and low placental passage rate, PTU has a lower impact on the foetus when compared to other commonly used drugs.

Second, we investigated the effects of PTU on neonatal hypothyroidism, which had not been analyzed in the previous meta-analysis. In animal studies, the current study found that PTU could cause foetal goitre [40]. Because propylthiouracil can suppress foetal thyroid function after the 11th post-conceptional week of pregnancy, when colloid begins to emerge in thyroid follicles and thyroxin can be detected, it should be avoided [41, 42]. Despite the theoretical link between PTU and foetal and neonatal thyroid hypothyroidism, our study found no significant difference. One possible mechanism is that the risk of neonatal hypothyroidism is significantly associated with the duration and cumulative dose of PTU, while the drugs used in the studies were clinically recommended doses. Propylthiouracil has no potential side effects for neonatal hypothyroidism or goitre if the dose is appropriately adjusted.

The current meta-analysis found that hyperthyroidism women treated with propylthiouracil had no significant differences in the ORs of hepatotoxicity compared to the control group, which was consistent with a previous study by Ping L et al. [43] Although PTU-induced liver disease was well documented in adults and children, there was limited data on hepatotoxicity in pregnant women and even less on effects on the foetus, resulting in conflicting opinions. Korelitz [15] discovered that women with hyperthyroidism who were not treated with PTU had higher rates of liver dysfunction than those who received PTU treatment. These findings could be explained by the link between untreated or inactive hyperthyroidism and elevated liver transaminases, or by the link between autoimmune disease and liver problems. There may be women whose pregnancy has been complicated by hyperthyroidism who are not being treated, and in these women, abnormalities in liver function tests are not uncommon, especially in women with hyperemesis gravidarum. It's also possible that doctors avoided giving antithyroid drugs to women who already had liver problems.

FT3, FT4, TT3 and TT4 are the first line parameters for diagnosis of clinical hyperthyroidism. These substances each have their own functions in the body and collectively reflect the thyroid gland's function. In our study of hyperthyroid women, TT3 and TT4 values were significantly lower in the treatment group compared to the control group. A recently published study lent support to this conclusion, claiming that propylthiouracil improved patient prognosis by controlling thyroid hormone levels in patients [43]. And FT3 and FT4 are hormones that bind to target cell receptors to perform their functions. Our study also analyzed the influence of PTU on FT3, FT4, and found that there was a significant difference in favour of PTU treatment. It is reasonable to expect that propylthiouracil can clearly relieve hyperthyroidism symptoms and improve thyroid gland function in pregnant patients with hyperthyroidism.

The current meta-analysis differed from others in that it assessed not only the influence of propylthiouracil in pregnant women with congenital anomalies, but also other effects of adverse pregnancy outcomes and its effectiveness.

## Limitations in current evidence

This meta-analysis has some limitations. First, we are unable to analyse specific birth defects because the included studies do not provide the original data. Equally important, children should be followed up after birth, and an agreement should be reached on the best time to follow up because some anomalies are not visible at birth. These factors may have influenced the results.

## Conclusions

In conclusion, the results of this meta-analysis for PTU treatment during pregnancy are reassuring because they show that it does not increase the risk of adverse pregnancy outcomes and can improve thyroid function in pregnant women with hyperthyroidism. Our evidence suggests that propylthiouracil is a safer and more effective option for treating pregnant women with hyperthyroidism. We believe our findings could be useful to physicians prescribing PTU to pregnant women with hyperthyroidism in the future, but more research is needed to confirm this conclusion.

## Supporting information

**S1 Materials. Search strategy.**
(DOCX)

**S1 Table. PRISMA checklist.**
(DOCX)

## Acknowledgments

The authors have no acknowledgments.

## Author Contributions

**Conceptualization:** Yiqun Miao, Aihua Wang.

**Data curation:** Yiqun Miao, Yang Xu, Ping Teng, Yuanyuan Zhang.

**Formal analysis:** Yiqun Miao, Yang Xu, Yun Zhou.

**Methodology:** Yiqun Miao, Aihua Wang, Yun Zhou, Wenwen Liu.

**Supervision:** Aihua Wang, Yuanyuan Zhang.

**Writing – original draft:** Yiqun Miao.

**Writing – review & editing:** Yiqun Miao, Aihua Wang, Yuanyuan Zhang.

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
