## [Decision Letter · Decision Letter 0]

25 Jan 2022

PONE-D-21-34654Efficacy of propylthiouracil in the treatment of pregnancy with hyperthyroidism and its effect on pregnancy outcomes A meta-analysisPLOS ONE

Dear Dr. Yiqun miao,

Thank you for submitting your manuscript to PLOS ONE. After careful consideration, we feel that it has merit but does not fully meet PLOS ONE’s publication criteria as it currently stands. Therefore, we invite you to submit a revised version of the manuscript that addresses the points raised during the review process.

We look forward to receiving your revised manuscript.

Kind regards,

Surasak Saokaew, PharmD, PhD, BPHCP, FACP, FCPA

Academic Editor

PLOS ONE

Journal Requirements:

3. PLOS requires an ORCID iD for the corresponding author in Editorial Manager on papers submitted after December 6th, 2016. Please ensure that you have an ORCID iD and that it is validated in Editorial Manager. To do this, go to ‘Update my Information’ (in the upper left-hand corner of the main menu), and click on the Fetch/Validate link next to the ORCID field. This will take you to the ORCID site and allow you to create a new iD or authenticate a pre-existing iD in Editorial Manager. Please see the following video for instructions on linking an ORCID iD to your Editorial Manager account: https://www.youtube.com/watch?v=_xcclfuvtxQ.

Reviewers' comments:

Reviewer's Responses to Questions

**Comments to the Author**

1. Is the manuscript technically sound, and do the data support the conclusions?

Reviewer #1: Yes

Reviewer #2: Yes

2. Has the statistical analysis been performed appropriately and rigorously? 

Reviewer #1: Yes

Reviewer #2: Yes

3. Have the authors made all data underlying the findings in their manuscript fully available?

Reviewer #1: Yes

Reviewer #2: No

4. Is the manuscript presented in an intelligible fashion and written in standard English?

Reviewer #1: Yes

Reviewer #2: No

5. Review Comments to the Author

Reviewer #1: This is a good-written manuscript presenting interesting research that aimed to finding outcome of efficacy and safety to use propylthiouracil during pregnancy with hyperthyroidism. Overall, methodology is valid and clearly. Studies were combinable and appropriate statistical methods used to combine results. Results and conclusion are generalizable.

Reviewer #2: The manuscript presents a meta-analysis that investigates the safety and efficacy of propylthiouracil during pregnancy. Results confirm the beneficial effects of propylthiouracil treatment. The risks of adverse pregnancy outcomes were not increased. PTU's efficacy in the treatment of pregnant women with hyperthyroidism was proven. The findings support the use of propylthiouracil during pregnancy with hyperthyroidism. This meta-analysis is important in order to summarize existing evidence and may provide clinical physicians with medication guidelines for pregnant women with hyperthyroidism. There are a few points to consider before publication can be recommended.

General: Improve spelling and mode of expression. There are several unfinished sentences and wrong phrases. In addition check whether correct English is used. Examples (not complete):

- people (abstract, better: patients, infants, pregnant women)

- sensitivity conducted analysis was conducted (methods)

- The study selection process is in Fig. 1. (results, better: is shown)

- propyliouracil instead of propylthiouracil (results, three times)

- A total of two literature were included in this study (results, better: studies)

- pregnant outcomes (discussion, better: pregnancy outcomes)

- in pregnant In this report (discussion, incomplete sentence)

- than those who were. (discussion, unfinished sentence)

- the treatment group significantly reduced the value of TT3, TT4 (discussion)

Introduction: "The evidence previously published research on the role of PTU treatment on the risk of pregnancy outcomes and its efficacy must be updated." Please check and improve this sentence.

Methods: "English language and Humans are limitations for searching"? What does this mean?

Intervention measures: "The experimental group received propylthiouracil treatment." The timing of treatment is important! The effect on birth defects can only be investigated when administered in the first trimester of pregnancy. Please add information on the timing of treatment from the original studies.

Exclusion criteria: "(4) Non-randomized controlled trials." What about cohort studies? Cohort studies were included in the analysis.

Description of bias risk may be wrong: "high risk" represents correct method or the complete data, illustrating small bias risk; "Unclear" means unclear method, illustrating moderate bias

risk; "low risk" indicates that incorrect method or incomplete data, illustrating high bias

risk. Other way round?

Supporting information

"Supplementary material 1. Search strategy." There is another supplementary material (not mentioned in this section).

Figure 1. Flowchart. Please add more numbers. What happened to 1017 records after checking for duplicates? Please add information on that. Please add numbers to the reasons. (-animal experiment (n=xx)).

6. PLOS authors have the option to publish the peer review history of their article (what does this mean?). If published, this will include your full peer review and any attached files.

Reviewer #1: **Yes: **Natthaya chaomuang

Reviewer #2: No

---

## [Author Response · Author response to Decision Letter 0]

16 Feb 2022

February 16, 2022

Response for manuscript PONE-D-21-34654 “Efficacy of propylthiouracil in the treatment of pregnancy with hyperthyroidism and its effect on pregnancy outcomes: A meta-analysis”

Dear Surasak Saokaew, PharmD, PhD, BPHCP, FACP, FCPA Academic Editors:

Thank you for providing us with such a great opportunity to submit a revised version of our manuscript. Meanwhile, we would like to express our sincere gratitude to all the reviewers for their detailed and constructive comments on our manuscript. According to those helpful suggestions, we have extensively revised the manuscript by correcting mistakes pointed out and supplemented the required materials to make our results convincing. At the same time, we ensure that the manuscript meets PLOS ONE's style requirements and has registered an ORCID iD of 0000-0002-6084-3662. All data, models, and code generated or used during the study appear in the submitted article.

We hope you will be satisfied with the revised version and look forward to hearing from you.

Sincerely,

The authors

Encl. Responses to the comments from Reviewer 1 and 2.

Reply to Reviewer #1

Dear Natthaya chaomuang,

Thank you very much for your time spent in reviewing our manuscript and for your encouraging comments on its merits. After careful consideration, we have further revised the article. We hope that you will be more satisfied with the revised version.

Comments:

“This is a good-written manuscript presenting interesting research that aimed to finding outcome of efficacy and safety to use propylthiouracil during pregnancy with hyperthyroidism. Overall, methodology is valid and clearly. Studies were combinable and appropriate statistical methods used to combine results. Results and conclusion are generalizable.”

Thank you very much for your affirmation of this article. We hope you will find this revised version more satisfactory. We are more than happy to make any further changes that will improve the article and facilitate successful publication.

Sincerely,

The Authors

Reply to Reviewer #2

Dear Reviewer,

We are really grateful to you for your time and efforts put into the comments. Those comments are all valuable and helpful for improving our manuscript.

In the remainder of this letter, we discuss each of your comments individually along with our corresponding responses. We appreciate your clear and detailed feedback and hope that the explanation has fully addressed all your concerns.

Comments:

“The manuscript presents a meta-analysis that investigates the safety and efficacy of propylthiouracil during pregnancy. Results confirm the beneficial effects of propylthiouracil treatment. The risks of adverse pregnancy outcomes were not increased. PTU's efficacy in the treatment of pregnant women with hyperthyroidism was proven. The findings support the use of propylthiouracil during pregnancy with hyperthyroidism. This meta-analysis is important in order to summarize existing evidence and may provide clinical physicians with medication guidelines for pregnant women with hyperthyroidism. There are a few points to consider before publication can be recommended.”

Thank you for your positive and valuable comments and hope our responses will exactly meet all your expectations.

To facilitate this discussion, we first retype your comments in italic font and then present our response to the comments.

Comment 1:

General: Improve spelling and mode of expression. There are several unfinished sentences and wrong phrases. In addition check whether correct English is used. Examples (not complete):

- people (abstract, better: patients, infants, pregnant women)

- sensitivity conducted analysis was conducted (methods)

- The study selection process is in Fig. 1. (results, better: is shown)

- propyliouracil instead of propylthiouracil (results, three times)

- A total of two literature were included in this study (results, better: studies)

- pregnant outcomes (discussion, better: pregnancy outcomes)

- in pregnant In this report (discussion, incomplete sentence)

- than those who were. (discussion, unfinished sentence)

- the treatment group significantly reduced the value of TT3, TT4 (discussion)

Response 1:

We really appreciate your suggestion. At the same time, we feel quite sorry and guilty that we have made such mistakes in spelling, grammar and expression. Thank you for your patience and understanding. Many thanks for your support! After careful examination, we have corrected those mistakes based on your helpful comments. We have also carefully reviewed the entire article to make sure that there is no other similar mistake.

-The researchers examined data from 13 randomized controlled trials and cohort studies involving 18948 infants. (The line number 16)

-If a study varied significantly from all other included studies in methods or results, a sensitivity analysis was used to exclude those studies from the meta-analysis. (The line number 152)

-Fig 1 shows the PRISMA flow chart for literature selection and the PRISMA checklist is shown in the S2. (The line number158)

-We have revised all propyliouracil to propylthiouracil.(The line number169, 195, 215)

-A total of two studies were included in this study. (The line number 232)

-We have revised all pregnant outcomes to pregnancy outcomes. (The line number185, 262, 265, 268)

-In this report, we utilized meta-analysis for the first time to summarize the pooled ORs of pregnancy outcomes and efficacy after exposure to propylthiouracil in pregnant women . (The line number 261)

-Korelitz[15] discovered that women with hyperthyroidism who were not treated with PTU had higher rates of liver dysfunction than those who received PTU treatment. (The line number 304)

-In our study of hyperthyroid women, TT3 and TT4 values were significantly lower in the treatment group compared to the control group. (The line number 314)

Comment 2:

Introduction: "The evidence previously published research on the role of PTU treatment on the risk of pregnancy outcomes and its efficacy must be updated." Please check and improve this sentence.

Response 2:

We feel great thanks for your kind comment. After much thought, we have changed this sentence to “Therefore, it is necessary to reassess previously published research evidence on the role of PTU treatment in the risk of pregnancy outcomes and its efficacy.” (The line number 53)

Comment 3:

Methods: "English language and Humans are limitations for searching"? What does this mean?

Response 3:

Thank you very much for pointing it out. In fact, our original intention is to include the English literature and exclude studies related to animal trial. According to your comment, We have changed this sentence to “Only studies involving humans were included for selection. Moreover, language limitations are English or Chinese.” (The line number 71)

Comment 4:

Intervention measures: "The experimental group received propylthiouracil treatment." The timing of treatment is important! The effect on birth defects can only be investigated when administered in the first trimester of pregnancy. Please add information on the timing of treatment from the original studies.

Response 4:

Your suggestion really mean a lot to us. We feel very sorry that we did not provide enough information about the timing of treatment before. According to your comment, we have added the duration of exposure of each study and it has been shown in table 1.

Comment 5:

Exclusion criteria: "(4) Non-randomized controlled trials." What about cohort studies? Cohort studies were included in the analysis.

Response 5:

Thank you for your nice suggestion. This research covers both RCT and cohort studies, so the exclusion criteria here have been changed to “(4)review, studies related to animal trial, in vitro study, case report and meeting abstract.” (The line number 108)

Comment 6:

Description of bias risk may be wrong: "high risk" represents correct method or the complete data, illustrating small bias risk; "Unclear" means unclear method, illustrating moderate bias

risk; "low risk" indicates that incorrect method or incomplete data, illustrating high bias

risk. Other way round?

Response 6:

Thank you for your reminding. We feel really sorry for this mistake. The sentence has been changed to‘"low risk" represents correct method or the complete data, illustrating small bias risk; "Unclear" means unclear method, illustrating moderate bias risk; "high risk" indicates that incorrect method or incomplete data, illustrating high bias risk.’ (The line number 132)

Comment 7:

Supporting information

"Supplementary material 1. Search strategy." There is another supplementary material (not mentioned in this section).

Response 7:

We are grateful to you for pointing it out. Actually, there was only one supplementary document, but in the process of revision and improvement, we added another one, and now there are two supplementary materials. One is search strategy, the other is PRISMA checklist. Therefore, we have changed this expression to "S1 materials. Search strategy." based on your comment. (The line number 158)

Comment 8:

Figure 1. Flowchart. Please add more numbers. What happened to 1017 records after checking for duplicates? Please add information on that. Please add numbers to the reasons. (-animal experiment (n=xx)).

Response 8:

Thanks for your nice suggestions. According to your comment, we have reworked the flow chart, and it is showed in Fig 1.

We would like to take this opportunity to thank you for all your time and efforts put into our manuscript, which is of great help to improve the quality of our manuscript. We hope you will be satisfied with our revised version.

We hope our revision covers all your concerns about the article. We would appreciate it if there are any other changes needing to be made to improve the quality of our article. We sincerely hope this article will be acceptable to be published on PLOS ONE. 

Sincerely, 

The Authors

---

## [Editor Report · Decision Letter 1]

23 Feb 2022

Efficacy of propylthiouracil in the treatment of pregnancy with hyperthyroidism and its effect on pregnancy outcomes: A meta-analysis

PONE-D-21-34654R1

Dear Dr.  Yiqun Miao,

We’re pleased to inform you that your manuscript has been judged scientifically suitable for publication and will be formally accepted for publication once it meets all outstanding technical requirements.

Kind regards,

Surasak Saokaew, PharmD, RPh, PhD, BPHCP, FACP, FCPA

Academic Editor

PLOS ONE
---

## [Editor Report · Acceptance letter]

28 Feb 2022

PONE-D-21-34654R1 

Efficacy of propylthiouracil in the treatment of pregnancy with hyperthyroidism and its effect on pregnancy outcomes:
A meta-analysis 

Dear Dr. Miao:

I'm pleased to inform you that your manuscript has been deemed suitable for publication in PLOS ONE. Congratulations! Your manuscript is now with our production department. 

Kind regards, 

on behalf of

Dr. Surasak Saokaew 

Academic Editor

PLOS ONE